# Causes of Post-Mortem Carcass and Organ Condemnations and Economic Loss Assessment in a Cattle Slaughterhouse

**DOI:** 10.3390/ani13213339

**Published:** 2023-10-27

**Authors:** Sebastian Ciui, Adriana Morar, Emil Tîrziu, Viorel Herman, Alexandra Ban-Cucerzan, Sebastian Alexandru Popa, Doru Morar, Mirela Imre, Adrian Olariu-Jurca, Kálmán Imre

**Affiliations:** 1Faculty of Veterinary Medicine, University of Life Sciences “King Mihai I” from Timişoara, 300645 Timișoara, Romania; sciui@yahoo.com (S.C.); emiltirziu@usvt.ro (E.T.); viorel.herman@fmvt.ro (V.H.); alexandracucerzan@usvt.ro (A.B.-C.); popa_sebastian_alexandru@yahoo.com (S.A.P.); dorumorar@usvt.ro (D.M.); mirela.imre@usvt.ro (M.I.); adrian.olariu-jurca@usvt.ro (A.O.-J.); 2Research Institute for Biosecurity and Bioengineering, University of Life Sciences ‘’King Mihai I” from Timișoara, 300645 Timișoara, Romania

**Keywords:** cattle, meat, slaughterhouse, lesions, financial loss

## Abstract

**Simple Summary:**

The inspection of edible parts from slaughtered animals, performed by veterinarians, is essential to ensure meat safety and to limit the exposure of consumers to risks. Likewise, this action constitutes a useful tool in providing baseline data for veterinary practitioners regarding the occurrence of disease in livestock at farm level, diseases which are not detectable in live animals. The detection of pathological lesions results in total or partial rejection of the meat, with subsequent economic losses for producers. The present study aimed to enhance the knowledge regarding these concerns by studying a total of 151,741 slaughtered cattle. The results showed that all types of edible parts of animals (carcasses, livers, lungs, hearts, and kidneys) presented one or more abnormalities, with various degrees of intensity, from 13.26% (for carcasses) to 0.27% (for kidneys). The overall financial losses due to carcass and organ condemnations was estimated at EUR 3,661,400.4 and EUR 360,316.9, respectively, representing 1.17% of the total achievable net revenue without rejections. The survey demonstrated that data recorded during meat inspections can be regarded as important sources of information for veterinary and public health authorities in terms of monitoring, control, and eradication programs for some diseases with public health and/or economic importance.

**Abstract:**

The study was undertaken to investigate the main causes of carcass and organ condemnations, as well as to estimate the financial losses suffered by a cattle slaughterhouse. In this regard, an active abattoir survey, based on standard post-mortem inspection procedures for meat, was conducted on 151,741 cattle, from January 2021 to December 2022. Overall, 13.27% (*n* = 20,125) of the carcasses expressed lesions or pathological conditions and, out of them, 1.15% (*n* = 1738) were totally confiscated, while another 12.12% (*n* = 18,387) were partially admitted for human consumption. In the case of organs, the general inspection data reveal that 12.28% (*n* = 18,630), 7.56% (*n* = 11,477), 1.89% (*n* = 2862), and 0.27% (*n* = 412) of the examined liver, lung, heart, and kidney specimens presented one or more types of abnormalities. In addition, regarding the types of specific pathological findings, dystrophies/anomalies (69.8%), circulatory disorders (40.6%), fecal contamination (60.9%), and suspected bacterial/viral infections showed a dominant occurrence in the liver, lung, heart, and kidneys, respectively. Consequently, the total direct financial losses resulting from edible part condemnation over the two years was estimated at EUR 4,021,717.3, which represents 1.17% of the total achievable net revenue without carcass and organ condemnation. Of this, EUR 3,661,400.4 (1.07%) and EUR 360,316.9 (8.73%) was related to carcass and organ condemnation, respectively. The study results demonstrate that the post-mortem inspection of meat at the slaughterhouse level plays a crucial role in identifying pathological lesions, in addition to some other issues, such as fecal contamination or non-compliant laboratory results, relevant to both public health and economic factors.

## 1. Introduction

The continuous increase in the human population should be accompanied by a proper growth rate in the production of foodstuffs of animal origin, in both industrialized and developing nations [1]. Nowadays, red meat (e.g., cattle, pigs, sheep, goats, and horses) is recognized as an important protein source in human nutrition. However, the continuous expansion of different infectious and parasitic-origin diseases, in the context of climate change, constitutes one of the greatest challenges for the meat industry in keeping the hierarchy of red meat in the human diet [2,3].

The availability of meat and products, thereof, in different parts of the world depends on many factors, including cultural habits, religious beliefs, and convenience. In this regard, beef is the third most widely consumed type of meat (~25%) [4], after pork and poultry, and is followed by sheep [5]. In 2020, in the European Union (EU), beef was regularly consumed by most citizens, and total production reached 6.8 million tons, the third highest production level worldwide, after the United States and Brazil. More than half of the EU’s beef is produced by three of its Member States, namely, France (21.2%), Germany (17.8%), and Italy (11.1%), with an annual per capita consumption of 10.3 kg [6].

Beef is widely recognized as containing several essential nutrients, including high-biological-value proteins, trace elements and vitamins [7], and it is regularly consumed by humans in most European countries. However, this type of meat has frequently been reported to harbor biological (e.g., pathogenic bacteria, viruses, or parasites) and chemical (e.g., residues of animal drugs, heavy metals, pesticides, mycotoxins, etc.) contaminants, resulting in a series of negative consequences for public health [8,9]. Thus, the safety and hygiene of meat intended for human consumption are fundamental issues for producers, distributors, and consumers to consider. To achieve this goal, the post-mortem inspection of meat at the slaughterhouse level, together with the control of residues of relevant substances, as introduced in Europe at the end of 19th century to protect public health, is of major importance [10]. The organoleptic post-mortem treatment of meat generally involves traditional techniques, including macroscopic inspection via palpation, incision, and olfaction, employing the faculties of sight, touch, and smell. These procedures are legislated in EU countries within the Official Controls Regulation (EU) no. 2017/625 and Commission Implementing Regulation (EU) no. 2019/627. They consist of the concise macroscopic anatomical–pathological examination of carcasses and organs after evisceration by an official, as well as an adequately trained veterinarian [11,12], followed by laboratory analyses for residues (e.g., antimicrobials), in order to ensure the safety of the meat and limit risks to the consumer. The obtained results provide baseline data regarding the occurrence of disease in livestock at the farm level, especially under subclinical conditions [1], and can be used in assessments of the economic loss related to organ and carcass condemnation. Moreover, meat inspection represents a last line of defense against some notable zoonotic diseases and is particularly important when planning preventive measures against consumer risks, as well as the control of disease at the farm level [3,10].

Epidemiological studies providing information on the occurrence and burdens of different cattle-specific diseases at the farm level have been undertaken in most countries around the world [13,14,15,16]. However, studies providing data on the extent to which bovine pathology can be inferred from macroscopic lesions at the slaughterhouse level, as well as associated evaluations of economic loss, remain limited [1,10]. Taking all this into consideration, the present study aimed to contribute to our knowledge on the main causes of carcass and organ condemnations, as well as to estimate the financial losses suffered by a cattle slaughterhouse.

## 2. Materials and Methods

### 2.1. Study Design

The study was conducted over a period of two years, focusing on 151,741 cattle slaughtered between January 2021 and December 2022, in a private slaughterhouse located in Bavaria (48.7904° N, 11.4979° E), southeast Germany. In this state, according to statistical data provided by www.fleischwirtschaft.de (accessed on 1 May 2023), there were approximately 2.94 million cattle in 2020, which is the highest number of any state nationwide [17].

Of the livestock regularly slaughtered in abattoirs (e.g., cattle, pigs, sheep, goats, and horses), cattle, and especially dairy cows, display the largest variations in disease history and diversity, due to their longer periods of exposure to different epidemiological factors throughout their productive cycle [10]. The abattoir we have selected collects animals to be slaughtered from industrial (from 100 to 10,000 heads), as well as small-scale integrated backyard, livestock production units (usually 20–30 heads), located within ~45,000 km^2^. The slaughtering capacity of the abattoir is around 330–350 cattle per day.

### 2.2. Meat Inspection, Data Collection, and Classification of the Lesions

All of the cattle enrolled in the present study arrived at the slaughtering unit accompanied by several official documents, including: (i) the health certificate required for live animals when they are transported from the holding unit to the slaughterhouse; (ii) a movement form; (iii) their food chain information; and (iv) their passport. In addition, the animals were slaughtered only if they passed the ante-mortem inspection. The aim of this step was to identify any indication of any condition that might negatively affect human or animal health. The post-mortem inspection of organs and carcasses was performed after the complete evisceration of the slaughtered animals, and was undertaken at two different points (one point for carcasses, and another for organs) by two official veterinarians (one of whom is an author of this study), following the procedure stipulated in Commission Implementing Regulation (EU) no. 2019/627 [12]. The meat examination procedure involved a systematic visual examination, as well as the palpation and olfaction, of the edible parts of the animals (e.g., carcass and visceral organs, including heart, lung, liver, and kidney), accompanied by the necessary incisions (e.g., various lymph nodes, the masseters muscle, heart, kidney, udder), in accordance with the requirements set out in the legislation [12]. Via the visual inspection, the general appearance of the organs and the muscle tissues, including their shape, color, dimensions, volume, smoothness, and the possible presence of abnormalities on their surface (e.g., nodules, abscesses, cysts, etc.), was evaluated. A palpation assessment, undertaken via touch with the application of pressure, was employed to evaluate the temperature, stiffness, elasticity, humidity, dryness, hardness, and friability of the structure of the muscle and viscera in order to detect abnormalities.

The post-mortem findings regarding macroscopic anatomical–pathological lesions in the carcasses and viscera of the slaughtered animals were registered on the slaughterhouse’s standardized VetScore^®^ (IMWT, GmbH, München, Germany) online platform. The definition, recognition, and classification of the pathological findings (Table 1) were carried out via the methods previously outlined by Gracey [18] and Vallant [19]. For the carcasses, the presence of generalized abnormalities (e.g., generalized diseases, changes such as a foreign smell, improper pH or jaundice, and non-compliant laboratory results) resulted in complete confiscation, while the presence of limited abnormalities (e.g., abscesses, feces contamination or the presence of dystrophies/anomalies) resulted in the affected parts being trimmed, with the remaining parts proceeding for human consumption. According to the policy of the slaughtering unit, with the exclusion of the spleen, which was immediately confiscated, organs proceeded for human consumption only if they were completely free of lesions. As regards the kidneys, if one presented lesions, both were judged unfit for human consumption. The types and causes of organ condemnation for each category are described in Table 1. In addition, data regarding the individual animals (e.g., age, gender) and their epidemiological background (e.g., farming system) are also listed.

### 2.3. Economic Loss Assessment

The estimation of the direct economic losses caused by the confiscation of edible offal and carcasses suffered by the slaughtering unit during the study period was carried out based on several variables, including (i) the total number of seized organs and carcasses, as well as their (ii) average weight (lung = 3.7 kg, liver = 6.5 kg, kidney = 0.5 kg, heart = 1.5 kg, and carcass = 340 kg for cows and 450 kg for bulls) [20,21], and (iii) the monetary value of one kilogram (in EUR, lung = 1, liver = 2.5, kidney = 2.3, heart = 3.3, carcass = 6) at the point of delivery from the slaughterhouse to the point of retail. In the case of partially confiscated carcasses, the estimated average loss, according to the statistical data reported on the slaughterhouse’s database, was 0.25%/carcass. Finally, an estimation of the total financial losses attributable to confiscation (expressed as a percentage) was obtained via the summation of the calculated losses for each type of organ and the carcasses, in relation to the total achievable net income without confiscations.

### 2.4. Statistical Analysis

The recorded data were statistically interpreted using the free online version of the GraphPad software (v 10.1.0) by Dotmatics^®^ (available online at: https://www.graphpad.com/quickcalcs/contingency1/, accessed on 1 July 2023). A chi-square (χ^2^) test with Yates’ correction, yielding a two-tailed *p* value, was used to compare differences in the pathological findings among the investigated organs and carcasses, in relation to the individual animal’s and the epidemiological data. Differences were considered significant at *p* ≤ 0.05.

## 3. Results

During the investigation, a total of 151,741 cattle were slaughtered and examined under sanitary veterinary conditions, using standard inspection procedures. The average number of animals slaughtered per working day was ~292. Overall, the post-mortem examinations showed that 1.15% (*n* = 1738; 95% CI = 1.1–1.2) of the carcasses were totally unfit for human consumption, while another 12.12% (*n* = 18,387; 95% CI = 11.96–12.29) were partially admitted for human consumption, resulting in a total of 13.27% (*n* = 20,125; 95% CI = 13.1–13.4) carrying lesions or pathological conditions. Regarding total carcass condemnations, the dominant causes were noted as abnormal changes (e.g., foreign smell, jaundice or non-compliant pH value), while dystrophies/anomalies were the most frequent causes of partial confiscations (Table 1).

Variable condemnation rates were identified amongst the examined organs (Table 1), with an overall prevalence of 22.0% (*n* = 33,388; 95% CI = 21.8–22.2) of them featuring lesions or pathological conditions. In detail, findings related to pathologic conditions were made in 12.28% (*n* = 18,630; 95% CI = 12.12–12.45), 7.56% (*n* = 11,477; 95% CI = 7.43–7.69), 1.89% (*n* = 2862; 95% CI = 1.82–1.96), and 0.27% (*n* = 412; 95% CI = 0.25–0.3) of the examined liver, lung, heart, and kidney specimens, respectively. In the livers, the most frequently encountered disorders were dystrophies/anomalies (69.8%), while circulatory disorders, contamination with feces, and suspected bacterial/viral infections were most dominant in the lungs (40.6%), hearts (60.9%), and kidneys (91.7%), respectively (Table 1). Across the study period, some categories of pathological findings were significantly more prevalent in 2021 (e.g., carcasses and lungs partially condemned due to fecal contamination, lungs condemned due to circulatory disorders, etc.) compared with 2022, while some were more prevalent in 2022 (e.g., carcasses totally condemned due to abnormal changes or suspected bacterial/viral lesions in the liver and kidney, etc.) (Table 1, indicated with *).

Table 2 summarizes the distribution of the findings following the post-mortem examinations of the slaughtered cattle, categorized via individual-level and epidemiological data. Notably, with regard to gender, all types of condemnation of carcasses and organs recorded, aside from fecal contamination, were statistically more prevalent (*p* < 0.05) in cows compared with bulls. The percentage of slaughtered cows and bulls was 67.59% and 32.26%, respectively. Similarly, except for parasitic and suspected bacterial/viral lesions in livers and hearts, respectively, as well as dystrophies/anomalies in kidneys, all the other causes of organ condemnation were significantly statistically more prevalent (*p* < 0.05) in cattle reared under intensive management systems, compared to those reared under extensive conditions (Table 2). Regarding the influence of age on the types of confiscations recorded, our results indicate positive statistical associations (*p* < 0.05) between cattle aged above three years and the presence of lesions (Table 2).

The direct financial loss estimations are summarized in Table 3. In relation to carcass confiscation, the total loss was estimated at EUR 3,661,400.4, representing 1.07% of the total achievable net revenue (EUR 341,752,320) without carcass condemnations. Likewise, the total loss due to organ seizures was calculated as EUR 360,316.9, representing 8.73% of the total achievable net income (EUR 4,127,355.2). The greatest losses were related to liver condemnations (EUR 302,737.5), and the smallest (EUR 947.6) to kidneys. Ultimately, at the investigated cattle slaughterhouse, the total financial loss associated with carcass and organ condemnations (EUR 4,021,717.8) stands at 1.16% of the achievable net income (EUR 345,879,675.2).

## 4. Discussion

The present investigation illustrates the utility of post-mortem inspections in monitoring bovine diseases leading to the condemnation of carcasses and organs, with financial implications for the meat industry. Moreover, the information obtained during these post-mortem inspections is useful in assessing consumer exposure to different zoonotic diseases. Further, at the end of this analysis, all of the parts of the slaughtered animals deemed edible must meet all of the sensory requirements for meat intended for human consumption, including in terms of the appearance on the surface and of sections, as well as the consistency, color, smell, and taste (evaluated only in heat-treated meat). If it does not meet the requirements, the meat and/or edible offal will be declared unfit for human consumption, even if some of the identified lesions or abnormalities do not have causes that could endanger public health.

In agreement with the aims of the present survey, several other studies, originating from different continents and countries, have been conducted in order to monitor the causes of carcass and organ condemnations, with extremely variable approaches [1,22,23,24,25,26,27,28,29,30,31,32]. Most of them processed data archived in the computerized databases of slaughtering units, and focused on specific types of parasitic, bacterial or viral and/or generalized lesions, with a special emphasis on the ramifications of hydatidosis and fasciolosis [1,24,25,31]. However, the number of studies focusing on the types of lesions encountered in the edible parts of slaughtered animals, as is the case in our survey, remains limited in the scientific literature [22,32]. Therefore, caution should be taken in comparing the results from the current study with those published by others, because some variables (e.g., study design, sample size, type of lesions addressed) can markedly influence the results.

In this context, a relevant study was conducted in Turkey by Yibar et al. [1] over a period of six months, in which a total of 5363 slaughtered cattle were subjected to post-mortem examinations using standard inspection procedures and considering a limited number of lesions. Accordingly, in the organs, the rate of condemnation due to hydatidosis and fasciolosis was 3.06% and 2.028%, respectively, while in carcasses, tuberculosis and jaundice accounted for 1.32% and 0.037% of cases deemed unfit for human consumption. In addition, in that study, a total of 2.33% of the offal and 0.32% of the carcasses were condemned, which are significantly lower values when compared with those of our survey (22.0% and 1.15%) [1]. A similar study was conducted in Tanzania by Tembo and Nonga [22], who reviewed a 3-year database on the causes of organ and carcass confiscation amongst the 85,980 cattle slaughtered in the Dodoma municipality and computed the resulting financial losses. A significantly lower number of carcasses with lesions (0.77%), and a lower rate of total rejection (0.05%), were recorded compared with our results (13.26% and 1.15%). On the contrary, the rates of occurrence of some conditions in the organs (30.2%) were higher compared with our study (22.0%), with a lower organ condemnation rate in the case of hearts (0.5% vs. 1.89%) and higher rates for livers (21.4% vs. 12.28%), lungs (10.5% vs. 7.56%), and kidneys (0.7% vs. 0.27%). Similarly, higher condemnation rates were identified amongst the organs inspected (e.g., liver—25.7%, lung—24.8%, kidney—0.5%, and heart—3.1%) in Ethiopia by Edo et al. [23]. Also, in a retrospective study (5-year period) conducted in Iran, Borji et al. [24] investigated the prevalence of parasitic infections resulting in viscera and carcass condemnations amongst the 125,593 cattle slaughtered in an abattoir in Ahwaz, Khuzestan Province. In this study, 7.9% of the livers, 3.45% of the lungs, and 0.01% of the carcasses were condemned due to parasitic infection, while these values in our study were 0.16%, 0.0%, and 0.0%, respectively. In another study conducted in an abattoir in Ismailia Governorate, Egypt, Ahmed et al. [25] compiled the results of post-mortem inspections of male cattle throughout a whole year. Regarding edible organs in bulls, 14.7% were judged unfit for human consumption, which is higher than the percentage we recorded (10.9%). Pathologically relevant findings were recorded in 44.6%, 21.3%, 17.9%, and 8.0% of the inspected lungs, kidneys, livers, and hearts, respectively, which are greater values than those from our study (5.07%, 0.02%, 4.39%, and 1.43%) for the same gender category (Table 2). Likewise, the proportions of lung (64.86%), liver (31.20%), heart (3.7%), and kidney (4.39%) condemnations in an Italian study undertaken by Ceccarelli et al. [26] over 7 years were significantly greater than those obtained in the current investigation [26]. The same trend was reported by Taha et al. [27] in South Sudan, whereby 26.1% and 17.2% of the liver and heart specimens they examined, respectively, were judged unfit for human consumption, while 58.6% of the 310 cattle carcasses examined yielded pathologically relevant findings, which is a considerably higher value compared to our study (13.26%).

As regards several types of lesions and the numbers of elements confiscated as a result, statistically significant differences have been found among the study years (Table 1). However, no scientific explanation can be offered, given the absence of any favorable, predisposing or determining factors related to these differences. According to our results, cows proved to be significantly more susceptible (*p* < 0.05) to pathological problems compared to bulls. This relation between gender and susceptibility to lesions can be partly explained by the much longer period of exposure among dairy cows to different pathogens or factors favoring disease throughout their productive cycle, as compared with bulls, which are generally slaughtered up to two years of age. Moreover, the differences identified in the management practices (e.g., the nature of the immunoprophylactic actions or the general welfare of the herd) and between the transmission dynamics of some diseases within dairy and fattening units should significantly influence the pathological findings [28].

Cattle reared under an intensive system exhibited significantly higher values (*p* < 0.05) for the majority of lesion types identified (Table 2), as compared with those managed under extensive farming systems. This is not surprising, as it is well known that animals reared within more confined production systems are particularly susceptible to production diseases [29]. In this regard, a series of predisposing, favoring or determining factors can be identified, such as the complex interactions between pathogens present within the farm, the presence of insufficient biosecurity measures, the increasing levels of animal stress and the disease pressures they face, and failures in management system quality (e.g., housing conditions, animal density or feeding) [28].

Notably, in the current study, cattle older than 3 years expressed significantly higher levels of the lesions identified in offal and carcasses compared to the younger ones (Table 2). This is not surprising given the repeated and longer exposure of older animals to different pathogens compared to younger ones, which can manifest in the high degree of pathology in this age category, as has been previously observed by several authors [1,7,10]. However, it is noteworthy that an extremely low prevalence of parasitic lesions was recorded in cattle younger than 3 years (7.2%) compared with mature animals (92.8%), given that young animals are apparently more susceptible to liver parasite infections due to their low immunity [30]. These results are in line with those published by Nyirenda et al. [31].

It is important to mention that the definitions of the majority of pathological findings (Table 1) made during the post-mortem inspections were dependent upon the knowledge, qualification, and experience of the veterinarians. Therefore, the final interpretation of macroscopic lesions is significantly determined by the subjective judgments of the veterinarians, and this represents a limitation to the present study. Thus, further investigations, processing the pathological post-mortem findings from a greater number of slaughterhouses, combined with etiological inferences related to lesions of parasitic and bacterial/viral origin, are still necessary in order to expand our knowledge about the causes of carcass and organ condemnations, and to evaluate the zoonotic risk faced by the consumer, along with the associated economic losses.

Overall, the total economic loss due to the condemnation of edible parts of slaughtered cattle was estimated in this study to be EUR 4,021,717.3, representing 1.17% of the total achievable net revenue without carcass and organ condemnation. Of this total, EUR 3,661,400.4 (91.04%) and 360,316.9 (8.96%) can be accounted for by carcass and organ condemnation, respectively. Regarding carcass examinations, condemnations due to abnormal changes (e.g., foreign smells, improper pH or jaundice) had the greatest financial impact, representing 51.68% (EUR 2,078,520) of the total losses. This finding implies that, within the post-mortem assessment, organoleptic examinations based on olfaction play an important role in identifying pathological conditions.

Several studies conducted in cattle abattoirs in different countries over the last few decades have reported estimations of monetary losses due to carcass and organ condemnation [1,22,23,24,25,27,32]. Most of these studies have presented details on the total economic losses resulting from the condemnation of carcasses and organs, expressing them in different monetary units, but do not provide data related to the breakdown of the damage to the total achievable net revenue. Thus, variable amounts of financial loss have been reported in Ethiopia (USD 202,688 for 23,064 slaughtered cattle) [32], Iran (USD 137,880 for 125,593 cattle, due to parasitic infections alone) [24], Sudan (USD 453,372 for 10,800 cattle) [27], and Turkey (USD 230,088 for 5353 cattle) [1]. Several factors can be taken into consideration when addressing the variations in the quantities of economic losses reported by different studies, including the number of animals slaughtered and the differences in management practices, the types of lesions considered and their prevalence, the items included in the analyses, the rates of rejection of organs, the duration of the study period, and the local market values of the edible parts of animals [1,23]. The financial losses related to pathological findings motivate improvements to the control strategies applied to the most common diseases in the areas of origin in slaughtered cattle.

## 5. Conclusions

The present investigation has demonstrated that a rigorous post-mortem inspection of the carcasses of slaughtered cattle, as part of the regulations on the so-called hygiene package, is essential to ensuring meat safety and limiting the risks to consumers. The survey also shows that carcass condemnations due to abnormal changes (e.g., foreign smells, improper pH or jaundice) (51.68% of the total financial losses from carcasses) and liver dystrophies/anomalies in the case of organ condemnations (58.64% of the total financial losses from organs) had the greatest financial impact. Further studies focusing on the etiology of pathological findings remain necessary in order to improve knowledge at the slaughterhouse level on the zoonotic risks imposed on the consumer.

## Figures and Tables

**Table 1 animals-13-03339-t001:** Overview of the main causes of carcass and organ condemnation in 151,714 cattle slaughtered at a private abattoir in Bavaria, Germany, over a two-year period.

Types and Causes of Carcass and Organ Condemnation	Year		Total (%)
	2021 (*n* = 80,117) (%)	2022 (*n* = 71,624) (%)	
Carcass			
Total			
generalized diseases (abscesses, tumors, septicemia)	323 (0.4)	298 (0.41)	621 (0.4%)
abnormal changes (smell, pH, jaundice)	422 (0.52)	593 * (0.82)	1015 (0.66)
non-compliant laboratory result (peritonitis, pleuritis, hemorrhagic diathesis)	48 (0.05)	54 (0.07)	102 (0.06)
Partial			
abscesses	2565 * (3.2)	1637 (2.28)	4202 (2.76)
feces contaminated	1728 * (2.15)	1180 (1.64)	2908 (1.91)
dystrophies/anomalies	4834 (6.03)	6443 * (8.99)	11,277 (7.43)
Lung			
suspected bacterial/viral or parasitic infections (pneumonia, abscesses, hydatidosis, lungworms, etc.)	1419 (1.77)	1358 (1.89)	2777 (1.83)
feces contamination	2794 * (3.48)	1245 (1.73)	4039 (2.66)
circulatory disorders (congestion, edema, emphysema, hemorrhages, pleurisy, etc.)	3076 * (3.83)	1585 (2.21)	4661 (3.07)
Liver			
parasitic (hydatidosis, fasciolosis, other trematodes, etc.)	142 (0.17)	108 (0.15)	250 (0.16)
suspected bacterial/viral (abscesses, perihepatitis)	1020 (1.27)	1207 * (1.68)	2227 (1.46)
feces contamination	1521 (1.89)	1629 * (2.27)	3150 (2.07)
dystrophies/anomalies (cirrhosis, icterus, hepatomegaly, etc.)	6480 (8.08)	6523 * (9.1)	13,003 (8.56)
Kidney			
suspected bacterial/viral (nephritis, necrosis)	173 (0.21)	205 * (0.28)	378 (0.24)
dystrophies/anomalies (hydronephrosis, congenital cysts, renal calculi, melanosis, etc.)	27 (0.03)	14 (0.01)	41 (0.02)
Heart			
suspected bacterial/viral (abscesses, pericarditis, endocarditis, etc.)	206 (0.25)	204 (0.28)	410 (0.27)
feces contamination	1112 * (1.38)	630 (0.87)	1742 (1.14)
dystrophies/anomalies (calcified cysts)	387 (0.48)	323 (0.45)	710 (0.46)

Legend: * statistically significant difference (*p* ≤ 0.05) between the number of confiscated elements recorded among the years.

**Table 2 animals-13-03339-t002:** Distribution of causes of carcass and organ condemnation in the slaughtered cattle, according to individual-level and epidemiological data.

Types and Causes of Carcass and Organ Condemnation	Total (*n* = 151,641)	Gender		Breeding System		Age	
		Cow (*n* = 102,543) (%)	Bull (*n* = 49,098) (%)	Intensive (*n* = 73,118) (%)	Extensive (*n* = 78,623) (%)	≤3 Year (*n* = 83,226) (%)	>3 Year (*n* = 68,415) (%)
Carcass							
Total							
generalized diseases	621	609 * (98.1)	12 (1.9)	420 * (67.6%)	201 (32.4)	43 (6.9)	578 * (93.1)
abnormal changes	1015	1003 * (98.8)	12 (1.2)	726 * (71.5)	289 (28.5)	38 (3.7)	977 * (96.3)
non-compliant laboratory result	102	98 * (96.1)	4 (3.9)	71 * (69.6)	31 (31.4)	12 (11.8)	90 * (88.2)
Partial							
abscess	4202	3838 * (91.3)	364 (8.7)	2572 * (61.2)	1630 (38.8)	517 (12.3)	3685 * (87.7)
feces contamination	2908	1589 * (54.6)	1319 (45.4)	1580 * (54.3)	1328 (45.7)	1267 (43.6)	1641 * (56.4)
dystrophies/anomalies	11,277	10,762 * (95.4)	515 (4.6)	7733 * (68.6)	3544 (31.4)	1498 (13.3)	9779 * (86.7)
Lung							
suspected bacterial/viral or parasitic	2777	2291 * (82.5)	486 (17.5)	1722 * (62.0)	1055 (38.0)	791 (28.5)	1986 * (71.5)
feces contamination	4039	2498 * (61.8)	1541 (38.2)	2234 * (55.3)	1805 (44.7)	1711 (42.4)	2328 * (57.6)
circulatory disorders	4661	4194 * (90.0)	467 (10.0)	2782 * (59.7)	1879 (40.3)	1028 (22.1)	3633 * (77.9)
Liver							
parasitic	250	250 (100)	-	126 (50.4)	124 (49.6)	18 (7.2)	232 (92.8)
suspected bacterial/viral	2227	2137 * (96.0)	90 (4.0)	1466 * (65.8)	761 (34.2)	131 (5.9)	2096 * (94.1)
feces contaminated	3150	1860 * (59.0)	1290 (41.0)	1729 * (54.9)	1421 (45.1)	1245 (39.5)	1905 * (60.5)
dystrophies/anomalies	13,003	12,224 * (94.0)	779 (6.0)	7997 * (61.5)	5006 (38.5)	1234 (9.5)	11,769 * (90.5)
Kidney							
suspected bacterial/viral	378	368 * (97.4)	10 (2.6)	206 * (54.5)	172 (45.5)	79 (20.9)	299 * (79.1)
dystrophies/anomalies	41	39 * (95.1)	2 (4.9)	22 (53.7)	19 (46.3)	2 (4.9)	39 * (95.1)
Heart							
suspected bacterial/viral	410	389 * (94.9)	21 (5.1)	217 (52.9)	193 (47.1)	56 (13.7)	354 * (86.3)
feces contamination	1742	1089 * (62.5)	653 (37.5)	888 * (51.0)	854 (49.0)	772 (44.3)	970 * (55.7)
dystrophies/anomalies	710	679 * (95.6)	31 (4.4)	401 * (56.5)	309 (43.5)	55 (7.7)	655 * (92.3)

Legend: * statistically significant differences (*p* ≤ 0.05) between the recorded number of confiscated elements between gender, breeding system, and age variables.

**Table 3 animals-13-03339-t003:** Estimated economic losses due to the condemnation of the edible parts of 151,741 slaughtered cattle (expressed in EUR).

Type of Carcass/Organs	No. of Condemned Carcasses/Organs	Average Weight (kg)	Total Condemned Weight (kg)	Price/kg (EUR)	Financial Losses (EUR) for the Study Period
Carcass					
Total					
cows	1710	340	581,400	6.00	3,488,400
bulls	28	450	12,600	6.00	75,600
Partial					
cows	16,189	340	13,760.65 *	6.00	82,563.9
bulls	2198	450	2472.75 *	6.00	14,836.5
Liver	18,630	6.5	121,095	2.5	302,737.5
Lung	11,477	3.7	42,464.9	1.0	42,464.9
Heart	2862	1.5	4293	3.3	14,166.9
Kidney	412	0.5 (×2)	412	2.3	947.6
Total					4,021,717.3

Legend: * the value is 0.25% of the total quantity of carcass condemnations, expressed in kg.

## Data Availability

The data are contained within the article.

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
