# Peer review of "Causes of Post-Mortem Carcass and Organ Condemnations and Economic Loss Assessment in a Cattle Slaughterhouse"

_animals, 2023, doi:10.3390/ani13213339_

Round 1

Reviewer 1 Report

Comments and Suggestions for Authors

Research assuring the quality of meat products from a farm to fork, involving the first stages of animal breeding as limiting in the quality of subsequent meat obtaining, a holistic study of the main reasons of condemnations is really important to have an overview of the state of the art of confiscations is of extreme importance. Most of the papers center in just one of the reasons that lead to the removal of carcasses or organs from the food chain at slaughterhouses levels. Nevertheless, the simultaneous consideration of several causes is of great novelty and pertinence. However, the study has to be greatly improved to be considered under publication. It lacks a lot information, the methodology is not well proposed and explained, discussion does not provide any information and conclusions are not supported by the findings of the research. Moreover, extensive editing English is required.

Lines 30-32. A little note that might be linked to calculations: The result of 1.15% + 12.12% is not 13.26%.

Lines 36-37: How can you distinguish a bacterial infection just by visual inspection?

Line 41: Please, revise the percentages.

Lines 42-44: The conclusions are not well expressed. The results show that the post mortem inspection at slaughterhouses level are linked to economical loses, but not only due to pathological lesions. You also describe some other issues such as fecal cross contamination that might be a cause of confiscation.

Lines 71-76: Not only the macroscopic anatomical-pathological examination by an official veterinarian is needed to assure the safety of meat and derived products. Also biological and chemical products such as antibiotics are included in official controls, and there is also a broad legislation landscape linked.

Line 93: Only bovines were slaughtered and revised?

Lines 107-108: There were no controls performed previous to slaughter in order to avoid unnecessary slaughters of animals that could be treated prior to slaughter?

Line 139: How did you calculated or obtained the economic value of the different organs?

Lines 146-149. Which program did you use to perform statistical analyses?

Table 1:
       Which were the objective considerations to assume that there wer generalized diseases or abnormal changes?
          Which were the laboratory analyses performed?
          How did you objectively evaluate the rest of the parameters?

Lines 183-184: Which was the percentage of bulls and cows slaughtered?

Lines 184 and onwards: Which is the limit for significant differences? p ≤ 0.05? p < 0.05?

Line 185: How did you know that the origin of the lesions was bacterial?

Tables 2 and 3: According to all data presented, which is the main etiological cause of economical loses?

Line numbering starts again in Discussion section.

Lines 5-7. This might be true but it is an issue that does not receive any attention during the rest of the study.

Line 25. This is study concretely reveals the origin of the lesions: tuberculosis. I miss a broader study of the etiological agents in order to increase the value of the results, to know whether meat might be a zoonotic agent, etc. Even more, to go one step behind and try to solve problems in the abattoirs prior to slaughter by implementing preventive measures.

Line 91: Where does this 1.07% comes from?

Lines 111-113: The authorities are not involved at any stage, only two veterinarians, but no other corrective measures apart from seizures are cited in the article. You refer at some point at legislation, but you don´t relate its content to the visual findings. Moreover, in lines 114-116 you conclude that the findings might help veterinarians in their clinical practice, but this is not a conclusion of the study. The aim of the study is to evaluate the visual events that lead to confiscation while veterinary inspection in one slaughterhouse. The next conclusion in lines 117-120 is a well-known result of seizures, and the one presented in the next sentence is well formulated, but it is a shallow issue easily extracted from any source. Please, rewrite the conclusions section to adapt them to your real findings.

Comments on the Quality of English Language

Use of English is not accurate and requires extensive editing.

Author Response

Reviewer≠1

Research assuring the quality of meat products from a farm to fork, involving the first stages of animal breeding as limiting in the quality of subsequent meat obtaining, a holistic study of the main reasons of condemnations is really important to have an overview of the state of the art of confiscations is of extreme importance. Most of the papers center in just one of the reasons that lead to the removal of carcasses or organs from the food chain at slaughterhouses levels. Nevertheless, the simultaneous consideration of several causes is of great novelty and pertinence. However, the study has to be greatly improved to be considered under publication. It lacks a lot information, the methodology is not well proposed and explained, discussion does not provide any information and conclusions are not supported by the findings of the research. Moreover, extensive editing English is required.

Dear Reviewer,

Special thanks for your efforts in reviewing our manuscript and your valuable comments which have greatly contributed to increasing its quality. We are delighted to read your words and we are grateful that our work to be considered for publication in the prestigious Animals journal!

Also, the paper has been edited by a professional MDPI English editing service under the identification number English editing ID: English-72478.

Please read below our answers to the raised concerns.

Lines 30-32. A little note that might be linked to calculations: The result of 1.15% + 12.12% is not 13.26%.

Answer: The authors apologize for this beginner mistake! The calculation was revised and corrected in 13.27%.

Lines 36-37: How can you distinguish a bacterial infection just by visual inspection?

Answer: The authors completely agree the reviewer opinion that a certain diagnosis in bacterial and/or viral infections can be established ONLY after laboratory investigations. Therefore, the raised concern was uniformly and carefully revised throughout the manuscript within a new formulation, resulting in – “suspected bacterial/viral infections” defining, in particular, localized infections (abscesses, phlegmon or other purulent collections) caused by pyogenic bacteria (staphylococci, streptococci, corynebacteria, pseudomonads) or lesions related by viruses.

Line 41: Please, revise the percentages.

Answer: Revised throughout the manuscript according to the reviewer suggestion, resulting in “Of this, EUR 3,661,400.4 (1.07%) and 360,316.9 (8.73%) were related to carcass and organ condemnation, respectively”

Lines 42-44: The conclusions are not well expressed. The results show that the post mortem inspection at slaughterhouses level are linked to economical loses, but not only due to pathological lesions. You also describe some other issues such as fecal cross contamination that might be a cause of confiscation.

Answer: Special thanks for this pertinent recommendation, the authors completely agree with this. The text was modified accordingly resulting in: “The study results demonstrate that the post mortem inspection of meat at the slaughterhouse level plays a crucial role in identifying pathological lesions, besides some other issues, such as fecal contamination or non-compliant laboratory results, relevant to both public health and economic factors.”

Lines 71-76: Not only the macroscopic anatomical-pathological examination by an official veterinarian is needed to assure the safety of meat and derived products. Also biological and chemical products such as antibiotics are included in official controls, and there is also a broad legislation landscape linked.

Answer: According to the reviewer recommendation, the text was rephrased/completed resulting in: “Thus, the safety and hygiene of meat intended for human consumption are fundamental issues for producers, distributors, and consumers to consider. To achieve this goal, the post mortem inspection of meat at the slaughterhouse level, together with the control of residues of relevant substances, as introduced in Europe at the end of 19th century to protect public health, is of major importance [10]. The organoleptic post mortem treatment of meat generally involves traditional techniques, including macroscopic inspection via palpation, incision, and olfaction, employing faculties of sight, touch, and smell. These procedures are legislated in EU countries within the Official Controls Regulation EU no. 2017/625 and the Commission Implementing Regulation (EU) no. 2019/627. They consist of the concise macroscopic anatomical–pathological examination of carcasses and organs after evisceration by an official as well as an adequately trained veterinarian [11,12], followed by laboratory analyses for residues (e.g., antimicrobials), in order to ensure the safety of the meat and limit risks for the consumer.”

Line 93: Only bovines were slaughtered and revised?

Answer: In the investigated slaughterhouse, besides cattle, pigs are also slaughtered. However, the present study focuses only on causes of post-mortem carcass and organ condemnations and economic loss assessment resulting from cattle slaughtering. In the near future, the research group will try to fulfill this approach by focusing, also, on pigs.

Lines 107-108: There were no controls performed previous to slaughter in order to avoid unnecessary slaughters of animals that could be treated prior to slaughter?

Answer: In order to clarify this concern, the following sentences were inserted in the revised version of the manuscript: “All of the cattle enrolled in the present study arrived at the slaughtering unit accompanied by several official documents, including: (i) the health certificate required for live animals when they are transported from the holding unit to the slaughterhouse; (ii) a movement form; (iii) their food chain information, and (iv) their passport. In addition, the animals were slaughtered only if they passed the ante mortem inspection. The aim of this step was to identify any indication of any condition that might negatively affect human or animal health”

Line 139: How did you calculated or obtained the economic value of the different organs?

Answer: The mentioned economic values of the edible parts of animals was obtained from the slaughterhouse database, representing their price at the delivery level from the slaughterhouse to retails.

Lines 146-149. Which program did you use to perform statistical analyses?

Answer: To clarify the raised concern, the following sentence was inserted in the revised version of the manuscript: The recorded data were statistically interpreted using the free online version of the GraphPad by Dotmatics® software (available online at: https://www.graphpad.com/quickcalcs/contingency1/).

Table 1:

Which were the objective considerations to assume that there were generalized diseases or abnormal changes?

Answer: The differentiation between the mentioned “generalized diseases” and “abnormal changes” categories was based on the fact that the first category can be linked by the occurrence of different infectious origin cattle diseases in subclinical conditions, which can pass the ante-mortem inspection barrier, while the second category is the results of abnormal meat changes after slaughtering or “surprise” after evisceration.

Which were the laboratory analyses performed?

Answer: The performed laboratory analyzes referred to the general assessment of the bacterial/viral contamination level of meat (e.g. determination of the total number of aerobic mesophilic germs, presence of coliform bacteria or the number of enterobacteria) or specific bacterial/viral analysis in case of suspecting the evolution of some diseases (e.g. leucosis, salmonellosis).

How did you objectively evaluate the rest of the parameters?

Answer: The authors addressed this concern by inserting the following sentences in the revised version of the manuscript. “It is important to mention that the definitions of the majority of pathological findings (Table 1) made during post mortem inspections were dependent upon the knowledge, qualification, and experience of the veterinarians. Therefore, the final interpretation of macroscopic lesions is significantly determined by the subjective judgments of the veterinarians, and this represents a limitation of the present study. Thus, further investigations, processing the pathological post mortem findings from a greater number of slaughterhouses, combined with etiological inferences related to lesions of parasitic and bacterial/viral origin, are still necessary in order to expand our knowledge about the causes of carcass and organ condemnations, and to evaluate the zoonotic risk faced by the consumer, along with associated economic losses.”

Lines 183-184: Which was the percentage of bulls and cows slaughtered?

Answer: The percentages of the slaughtered cows and bulls were 67.59% and 32.26%, respectively.

Lines 184 and onwards: Which is the limit for significant differences? p ≤ 0.05? p < 0.05?

Answer: As the authors mentioned within the statistical analysis of the obtained data, differences were considered significant at p ≤ 0.05, meaning recorded values greater than or at least equal to 0.05.

Line 185: How did you know that the origin of the lesions was bacterial?

Answer: as the authors previously mentioned, the formulation of “bacterial origin lesions” was rephrased throughout the manuscript resulting in “suspected bacterial/viral origin lesions”.

Tables 2 and 3: According to all data presented, which is the main etiological cause of economical loses?

Answer: The main etiological cause of economical loses was the total carcass condemnation due to abnormal changes like foreign smell, improperly pH or jaundice, which represent 51.68% (2,078,520 €) out from the total economic losses. In case of organs, the liver condemnations due to dystrophies/anomalies has the largest financial impact. These data was introduced in the revised version of Conclusions resulting in: “The survey also shows that condemnations due to abnormal changes (e.g., foreign smells, improper pH, or jaundice) in the case of carcasses (51.68% of the total financial losses from carcasses) and liver dystrophies/anomalies in the case of organs (58.64% of the total financial losses from organs) had the greatest financial impact.”

Line numbering starts again in Discussion section.

Answer: this issue was revised in the new version of the manuscript.

Lines 5-7. This might be true but it is an issue that does not receive any attention during the rest of the study.

Answer: The authors agree on the reviewer opinion. Therefore, the sentence was rephrased resulting in: “The present investigation illustrates the utility of post mortem inspections in monitoring bovine diseases leading to the condemnation of carcasses and organs, with financial implications for the meat industry.”

Line 25. This is study concretely reveals the origin of the lesions: tuberculosis. I miss a broader study of the etiological agents in order to increase the value of the results, to know whether meat might be a zoonotic agent, etc. Even more, to go one step behind and try to solve problems in the abattoirs prior to slaughter by implementing preventive measures.

Answer: The authors were to underline the fact that the aim of the present study was to evaluate the main causes of carcass and organ condemnations and to estimate the associated financial losses, without providing data on the involvement of any etiological agents in the accounted lesions. This will be the aim of another study within the Ph.D. thesis of the first author. However, this issue was mentioned as a limitation of the present investigation. Likewise, according to the regulations in force, the health certificate for live animals transported from the holding to the slaughterhouse and the food chain information documents provided by the veterinarian responsible for the management of herd health status, must certify that animals are healthy and suitable for slaughtering. As the authors mentioned in the Conclusion section of the submitted manuscript the recorded pathological findings for carcasses and offal during post-mortem inspection provide baseline data for primary production units (farmers) about the occurrence of different types of disease, which can greatly contribute to the enhancement of herd health management strategies by veterinarians.

Line 91: Where does this 1.07% comes from?

Answer: The value 1.07% represent the computed total and associated proportional losses (3,661,400.4 €), reported to the total amount of the achievable net revenue (341,752,320 €) and without condemnations in case of carcasses.

Conclusions

Lines 111-113: The authorities are not involved at any stage, only two veterinarians, but no other corrective measures apart from seizures are cited in the article. You refer at some point at legislation, but you don´t relate its content to the visual findings. Moreover, in lines 114-116 you conclude that the findings might help veterinarians in their clinical practice, but this is not a conclusion of the study. The aim of the study is to evaluate the visual events that lead to confiscation while veterinary inspection in one slaughterhouse. The next conclusion in lines 117-120 is a well-known result of seizures, and the one presented in the next sentence is well formulated, but it is a shallow issue easily extracted from any source. Please, rewrite the conclusions section to adapt them to your real findings.

Answer: according to the reviewer request, the conclusion section was almost completely rewritten resulting in:

“The present investigation has demonstrated that a rigorous post mortem inspection of the carcasses of slaughtered cattle, as part of the regulations of the so-called hygiene package, is essential to ensuring meat safety and limiting risks for consumers. The survey also shows that condemnations due to abnormal changes (e.g., foreign smells, improper pH, or jaundice) in the case of carcasses (51.68% of the total financial losses from carcasses) and liver dystrophies/anomalies in the case of organs (58.64% of the total financial losses from organs) had the greatest financial impact. Further studies focusing on the etiology of pathological findings remain necessary in order to improve knowledge at the slaughterhouse level of the zoonotic risks imposed on the consumer.”

 THANK YOU AGAIN!

Reviewer 2 Report

Comments and Suggestions for Authors

This study demonstrated the  causes of post-mortem carcass and organs condemnations and economic loss assessment in a cattle slaughterhouse
 which is fundamental in ensuring meat safety and to limit risks for the
consumer.  And this study was well designed and written. I just have several quesitons and suggestions below:

1 line 108, which were the two different points? Please show it clearly.

2 the legend * in all tables were not clearly defined or explained. For example, what did the difference mean in Table 1? between different years? or else? It should be clearly delared below the table.

3 Why did you only showed 0.25% of the total amount of condemnations in Table 3? How about the other data in this table? Was it also 0.25% or the total amount? It was so confused.

4 It would be much better to comapre the differents between carcass and organs in table 1 and table 3, and to explain or discuss the causes that made the differents between them.

5 A bit regretful for this study that it just showed the investigation results in only one cattle slaughterhouse, it would be much helpful and persuasive if the investigation could be expanded to more slaughterhouses in different regions in future.

Author Response

Reviewer≠2

This study demonstrated the causes of post-mortem carcass and organs condemnations and economic loss assessment in a cattle slaughterhouse which is fundamental in ensuring meat safety and to limit risks for the consumer. And this study was well-designed and written. I just have several questions and suggestions below:

Dear reviewer,

Special thanks for your efforts in reviewing our manuscript and your valuable comments and questions to the quality of our submission. We are delighted to read your words and we are very happy to be considered for publication in the prestigious Animals journal! Please read below our answers to the raised concerns.

  1. line 108, which were the two different points? Please show it clearly.

Answer: The authors modified the sentence resulting in: “The post mortem inspection of organs and carcasses was performed after the complete evisceration of the slaughtered animals, and was undertaken in two different points (one point for carcasses, and another for organs) by two official veterinarians (one of whom was an author of this study), following the procedure stipulated in the Commission Implementing Regulation (EU) no. 2019/627 [12].”

  1. the legend * in all tables were not clearly defined or explained. For example, what did the difference mean in Table 1? between different years? or else? It should be clearly delared below the table.

Answer: The authors completed the legend of Tables with:

Table 1 - “Legend: *—statistically significant difference (p ≤ 0.05) between the numbers of confiscated elements recorded among years.”

Table 2 – “Legend: *—statistically significant differences (p ≤ 0.05) between the recorded numbers of confiscated elements between gender, breeding system, and age variables.”

  1. Why did you only showed 0.25% of the total amount of condemnations in Table 3? How about the other data in this table? Was it also 0.25% or the total amount? It was so confused.

Answer: The authors were to strengthen the fact that the mentioned 0.25% in Table 3 refers ONLY to partial condemnations in the case of carcasses and according to gender, as was mentioned in lines 139-140 of the original submission “In the case of partially confiscated carcasses, the estimated average loss, according to the statistical data reported on the slaughterhouse’s database, was 0.25%/carcass.” However, the layout of the Table 3, probably, created a misunderstanding, for which the authors apologize! The authors hope that the new design of Table 3 can clarify this confusion!

  1. It would be much better to comapre the differents between carcass and organs in table 1 and table 3, and to explain or discuss the causes that made the differents between them.

Answer: With respect to the reviewer recommendation, in the author’s opinion the suggested comparisons are not scientifically sound, because there are diseases in which the pathological findings are present only at the organ level, and others, less often, only on the carcasses.

5 A bit regretful for this study that it just showed the investigation results in only one cattle slaughterhouse, it would be much helpful and persuasive if the investigation could be expanded to more slaughterhouses in different regions in the future.

Answer: The authors completely agree with the reviewer opinion. The authors want to highlight the fact that this study was carried out in agreement with the planned actions within a Ph.D. thesis of the first author entitled “Inspection and official control of red meat in abattoirs: contributions to the improvement of meat quality and safety”. Recently, the authors received approval from another slaughtering unit for a new study, having a new opportunity to establish correlations within the influence of epidemiological (environmental and animal-related) factors on meat quality, and implicitly on economic losses due to carcass and organ condemnation.

Thank you again!

Reviewer 3 Report

Comments and Suggestions for Authors

The paper summarize the cause of meat confiscations in slaughterhouses and its economical consequences. In reviewer opinion its economical aspect of work improves value of paper. In did, there are few papers describing cost looses for cattle industry.

The main weakness of work is poor description of the economical looses. Authors focused only one simple calculation of costs connected with total partial confirmations of meat for all industry. I would suggest to extend the calculation of extra aspects f.e. cost looses depending on size of farm (cost for farm), cost looses per one animal etc. It would improve the importance of economical value of paper. I would recomended authors to considered that point of view.  

List of necessary corrections:

Please pay attention of writing post-mortem in italic, be consequent: line 25, line 154, etc.

line 92: Study design, please give more details how investigated population was selected? Randomly? Please indicate the number of animals from small and big farms, size of batches, age of animals as long as in Discussion section authors mentioned that those conditions influenced on result of slaughterhouse examination (page 8 line 65)

linÄ™ 138 pay attention to marks in brackets etc., lack of Euro mark next to liver

linÄ™ 162 and 175 both sentences give the same information, dubbing is not necessary, delete one of them

Table 1 please pay attention to table layout and be consequent in designing all tables. It simplifies to reader following the data. In Table one word Totally (verse 3) is centered while word partially is on the left, both indicates type of meat confiscates so should be in the same level, please precise this

Table 1 Are statistical data really necessary as long as statistically significant difference between number of confiscated elements among years dose not give neither valuable informations nor conclusions. In Discussion sections there is no divagations of reason of this difference (epidemiological status, housekeeping conditions, etc.)

In Discusion sections the line numbering starts again

page 7 line 12 please provide list of references confirming the statement in sentence

page 7 line 15 the same comment as above

page 7 line 17 the same comment

Author Response

Reviewer≠3

The paper summarize the cause of meat confiscations in slaughterhouses and its economical consequences. In reviewer opinion its economical aspect of work improves value of paper. In did, there are few papers describing cost looses for cattle industry.

Dear reviewer,

Special thanks for your efforts in reviewing our manuscript and your valuable comments and questions to the quality of our submission. We are delighted reading your words and we are very happy to be considered for publication in the prestigious Animals journal! Please read below our answers to the raised concerns.

The main weakness of work is poor description of the economical looses. Authors focused only one simple calculation of costs connected with total partial confirmations of meat for all industry. I would suggest to extend the calculation of extra aspects f.e. cost looses depending on size of farm (cost for farm), cost looses per one animal etc. It would improve the importance of economical value of paper. I would recomended authors to considered that point of view.

Thank you for this pertinent suggestion! With respect to the reviewer recommendation, the authors want to highlight the fact that this study was carried out in agreement with the planned actions within a PhD thesis of the first author entitled “Inspection and official control of red meat in abattoirs: contributions to the improvement of meat quality and safety”. According with the approved working plan of the PhD student and in agreement with the reviewer suggestion, another manuscript is under preparation, aiming to establish correlations within the influence of epidemiological (environmental and animal related) factors on meat quality, and implicitly on economic losses due to carcass and organs condemnation. The authors express their desire that the purpose of this study be a priority for the scientific community, and without any overlapping of with the results of the present work.

List of necessary corrections:

Please pay attention of writing post-mortem in italic, be consequent: line 25, line 154, etc.

Answer: Modified throughout the manuscript according to the reviewer suggestion.

line 92: Study design, please give more details how investigated population was selected? Randomly? Please indicate the number of animals from small and big farms, size of batches, age of animals as long as in Discussion section authors mentioned that those conditions influenced on result of slaughterhouse examination (page 8 line 65)

Answer: The manuscript processes all the data obtained from the sanitary-veterinary post-mortem inspection of the all cattle admitted for slaughtering in the unit. The abattoir slaughters animals either on the basis of a previously established contract, or on the basis of requests by the breeders. So, in this regard the random selection is not suitable. The requested data about the breeding units size was completed thus “The abattoir we have selected collects animals to be slaughtered from industrial (from 100 to 10,000 heads) as well as small-scale integrated backyard livestock production units (usually 20-30 heads), located within ~120 km2.” Two age categories was taken in consideration (≤ 3 year vs. >3 year) which are mentioned in the Table 2.

linÄ™ 138 pay attention to marks in brackets etc., lack of Euro mark next to liver

Answer: Thank you for the suggestion, corrected!

linÄ™ 162 and 175 both sentences give the same information, dubbing is not necessary, delete one of them

Answer: According to the reviewer recommendation, the lines 175-177 were deleted.

Table 1 please pay attention to table layout and be consequent in designing all tables. It simplifies to reader following the data. In Table one word Totally (verse 3) is centered while word partially is on the left, both indicates type of meat confiscates so should be in the same level, please precise this

Answer: Thank you for the precise indication. The design of all tables was revised and uniformed. Corrected as recommended!

Table 1 Are statistical data really necessary as long as statistically significant difference between number of confiscated elements among years dose not give neither valuable informations nor conclusions. In Discussion sections there is no divagations of reason of this difference (epidemiological status, housekeeping conditions, etc.)

Answer: In order to be consequent in the data presentation and interpretation between the Table 1 and Table 2, the authors would like to keep the symbols (*) indicating statistically significant differences. However, to complete this answer, the following sentence was inserted in the manuscript: “As regards several types of lesions and the numbers of elements confiscated as a result, statistically significant differences have been found among the study years (Table 1). However, no scientific explanation can be offered, given the absence of any favorable, predisposing or determining factors related to these differences”

In Discusion sections the line numbering starts again

Answer: Revised!

page 7 line 12 please provide list of references confirming the statement in sentence

Answer: The references were inserted as requested.

page 7 line 15 the same comment as above

Answer: The references were inserted as requested.

page 7 line 17 the same comment

Answer: The references were inserted as requested.

Thank you again!

Round 2

Reviewer 1 Report

Comments and Suggestions for Authors

Thanks for considering all the changes suggested.

Author Response

Thank you again for your time and efforts in reviewing our manuscript, and your final approval for considering it as publication!